# A G1528C *Hadha* knock-in mouse model recapitulates aspects of human clinical phenotypes for long-chain 3-hydroxyacyl-CoA dehydrogenase deficiency

Garen Gaston[1], Shannon Babcock [1], Renee Ryals[1,2], Gabriela Elizondo [1], Tiffany DeVine[1], Dahlia Wafai [2], William Packwood[3], Sarah Holden[4], Jacob Raber [3,4,5,6], Jonathan R. Lindner[3,7], Mark E. Pennesi[2], Cary O. Harding [1] & Melanie B. Gillingham [1✉]

Long chain 3-hydroxyacyl-CoA dehydrogenase deficiency (LCHADD) is a fatty acid oxidation disorder (FAOD) caused by a pathogenic variant, c.1528 G > C, in *HADHA* encoding the alpha subunit of trifunctional protein (TFPα). Individuals with LCHADD develop chorioretinopathy and peripheral neuropathy not observed in other FAODs in addition to the more ubiquitous symptoms of hypoketotic hypoglycemia, rhabdomyolysis and cardiomyopathy. We report a CRISPR/Cas9 generated knock-in murine model of G1528C in *Hadha* that recapitulates aspects of the human LCHADD phenotype. Homozygous pups are less numerous than expected from Mendelian probability, but survivors exhibit similar viability with wildtype (WT) littermates. Tissues of LCHADD homozygotes express TFPα protein, but LCHADD mice oxidize less fat and accumulate plasma 3-hydroxyacylcarnitines compared to WT mice. LCHADD mice exhibit lower ketones with fasting, exhaust earlier during treadmill exercise and develop a dilated cardiomyopathy compared to WT mice. In addition, LCHADD mice exhibit decreased visual performance, decreased cone function, and disruption of retinal pigment epithelium. Neurological function is affected, with impaired motor function during wire hang test and reduced open field activity. The G1528C knock-in mouse exhibits a phenotype similar to that observed in human patients; this model will be useful to explore pathophysiology and treatments for LCHADD in the future.

[1] Department of Molecular and Medical Genetics, Oregon Health and Science University, Portland, OR, USA. [2] Casey Eye Institute, Oregon Health and Science University, Portland, OR, USA. [3] Knight Cardiovascular Institute, Oregon Health and Science University, Portland, OR, USA. [4] Department of Behavioral Neuroscience, Oregon Health and Science University, Portland, OR, USA. [5] Departments of Neurology and Radiation Medicine, Oregon Health and Science University, Portland, OR, USA. [6] Division of Neuroscience, Oregon National Primate Research Center (ONPRC), Oregon Health and Science University, Portland, OR, USA. [7] Present address: Cardiovascular Division, University of Virginia Medical Center, Charlottesville, VA, USA. ✉email: gillingm@ohsu.edu

Long-chain 3-hydroxyacyl CoA dehydrogenase deficiency (LCHADD, OMIM# 609016) is a rare, recessively inherited disorder of fatty acid oxidation (FAO) caused by the presence of a common pathogenic variant, c.1528 G > C, in at least one allele of the *HADHA* gene[1,2]. *HADHA* encodes the alpha subunit of mitochondrial trifunctional protein (TFPα) that together with the beta subunit (TFPβ), encoded by *HADHB*, form a heterotetrameric multi-functional complex catalyzing the last three steps in mitochondrial long-chain fatty acid β-oxidation: long-chain enoyl CoA hydratase, long-chain 3-hydroxyacyl CoA dehydrogenase (LCHAD) and long-chain ketothiolase (Supplementary Fig. 1a)[3]. The common LCHADD pathogenic variant results in a glutamic acid to glutamine amino acid change (E510Q) in the LCHAD active site of TFPα reducing its activity but leaving the other two enzymatic activities relatively intact[1]. The biochemical consequence is reduced FAO capacity with activation of alternate energy metabolism pathways and the accumulation of partially oxidized plasma 3-hydroxy-acylcarnitines. In contrast, global trifunctional protein deficiency (TFPD, OMIM #609015) is caused by other less common variants in either the *HADHA* or *HADHB* genes that lead to decreased protein stability and loss of all three enzymatic functions[4]. Similar to other long-chain FAO disorders, humans with LCHADD or TFPD frequently present in infancy with hypoketotic hypoglycemia, liver dysfunction, and dilated cardiomyopathy, but then develop recurrent rhabdomyolysis later in life[5]. However, several disease complications are unique to disorders impacting mitochondrial TFP activity including peripheral axonal sensorimotor neuropathy and a progressive retinopathy with vision loss. Patients with LCHADD and TFPD have similar peripheral neuropathy characterized by relatively ubiquitous infantile or childhood loss of peripheral deep tendon reflexes with variable progression to foot drop, numbness, muscle weakness and loss of mobility that waxes and wanes over time[6,7]. In contrast, retinopathy, characterized by a steady decline in visual acuity, increased myopia, and progressive atrophy of the outer retina with choriocapillaris loss, appears to progress more rapidly in patients with LCHADD[8].

Multiple knockout mouse models have been created for studying FAO disorders; however, a viable mouse model that recapitulates LCHADD did not exist. Two prior TFPα knockout models resulted in neonatal lethality in homozygous pups[9,10]. Because there has not been a mouse model that successfully recapitulates the human disorder, we sought to use CRISPR/Cas9 gene editing technology to introduce the common LCHADD pathogenic variant into mouse embryos and create a model that could be used to study LCHADD in vivo. Here we report a LCHADD mouse that exhibits features of the human disorder including impaired fatty acid oxidation, accumulation of 3-hydroxy-acylcarnitines, low fasting ketone concentrations, impaired motor function, exercise intolerance, and dilated cardiomyopathy. They also develop a retinal phenotype with decreased visual performance.

## Results

### Generation of G1528C LCHADD mice. 
Similar to humans, the murine c.1528 G nucleotide is located in *Hadha* on exon 15 and in the same codon position of amino acid 510, which is in a stretch highly conserved at the amino acid level. Using the CRISPR/Cas9 system, the c.1528 G to C (GAA to CAA) mutation was introduced into exon 15 by homology-directed repair in fertilized C57Bl/6 J embryos, and a silent mutation (ACC to ACG) was introduced to the PAM sequence to prevent the binding and re-cutting of the edited allele by the Cas9 system post-repair (Fig. 1a). The G1528C sequence results in a glutamic

acid to glutamine amino acid change (E510Q) in the LCHAD active site of the protein identical to that observed in human patients with LCHADD (Supplementary Fig. 1b). Embryos treated with editing reagents were implanted into pseudo-pregnant females and allowed to develop to term. Progeny were then screened for the presence of the c.1528 G > C variant by allele-specific polymerase chain reaction (PCR) on tail biopsy DNA to detect heterozygotes. Homozygous mice were then generated by mating heterozygous breeding pairs. The line was further propagated by mating LCHADD homozygous males with heterozygous females. Week old homozygous mouse pups were identified at ~50% lower frequency than expected regardless of breeding approach, suggesting reduced LCHADD mouse pup viability similar to observations in the long-chain acyl-CoA dehydrogenase (LCAD) knockout mouse (Table 1)[11,12]. However, after weaning, surviving homozygous LCHADD mice grew normally with similar body and liver weights to wild-type (WT) littermates (Supplementary Fig. 2a, b). Hereafter, the terminology LCHADD mice will refer to mice homozygous for the *Hadha* c.1528 G > C allele.

Normal levels of TFPα have been detected in fibroblasts of human patients homozygous for c.1528 G > C so we anticipated normal protein expression in LCHADD mouse tissues[1,4]. We analyzed heart and liver tissue for protein levels of TFPα, TFPβ and very long-chain acyl-CoA dehydrogenase (VLCAD), the FAO pathway enzyme immediately proximal to mitochondrial trifunctional protein. Protein bands were normalized to GAPDH and compared between genotypes by sex. Similar levels of TFPα, TFPβ and VLCAD protein were detected in heart of male and female WT and LCHADD mice (Fig. 1b, c, Supplementary Fig. 6). VLCAD protein in liver was similar between genotypes. However, TFPα and TFPβ was reduced in male and female LCHADD liver compared to sex-matched WT liver (Fig. 1b, c, Supplementary Fig. 6). To determine if decreased protein levels were related to decreased mRNA levels, we analyzed gene expression of *Hadha* and *Hadhb* by reverse transcription-quantitative PCR (RT-qPCR). We found similar *Hadha* and *Hadhb* mRNA expression in both heart and liver of male and female LCHADD and WT mice (Fig. 1d and Supplementary Fig. 2c). Decreased TFPα and TFPβ protein levels but similar gene expression in liver of LCHADD mice suggests decreased protein stability and/or increased turnover of TFPα/TFPβ in the liver.

To determine the impact of the G1528C mutation on FAO in tissues, palmitate oxidation studies were performed on liver and cardiac tissue. Fresh tissue homogenates were incubated with $^{14}C$-palmitate for 30 min then measurements were taken for the radiolabeled carbon dioxide ($^{14}CO_2$) production and residual radioactivity in acid soluble products. Both liver and cardiac tissue exhibited lower metabolized $^{14}C$ compared to those seen in liver and cardiac tissue of WT mice (Fig. 1e). Palmitate oxidation was approximately 60% of WT suggesting reduced but not absent FAO capacity. A 60% residual palmitate oxidation seems relatively high for LCHADD. However, our assay measures total metabolized $^{14}C$ including β-oxidation in the peroxisome and w-oxidation in microsomes. Peroxisomal oxidation may compensate for lower mitochondrial oxidation in tissues with impaired mitochondrial β-oxidation and may explain higher than anticipated tissue palmitate oxidation[13]. Regardless, total metabolized $^{14}C$ was significantly lower in the LCHADD tissues compared to WT mice.

### Male LCHADD mice oxidize less fat and more glucose than WT mice. 
Indirect calorimetry was used to determine whole body substrate oxidation. The volumes of consumed oxygen and released carbon dioxide were analyzed in individual mice in a

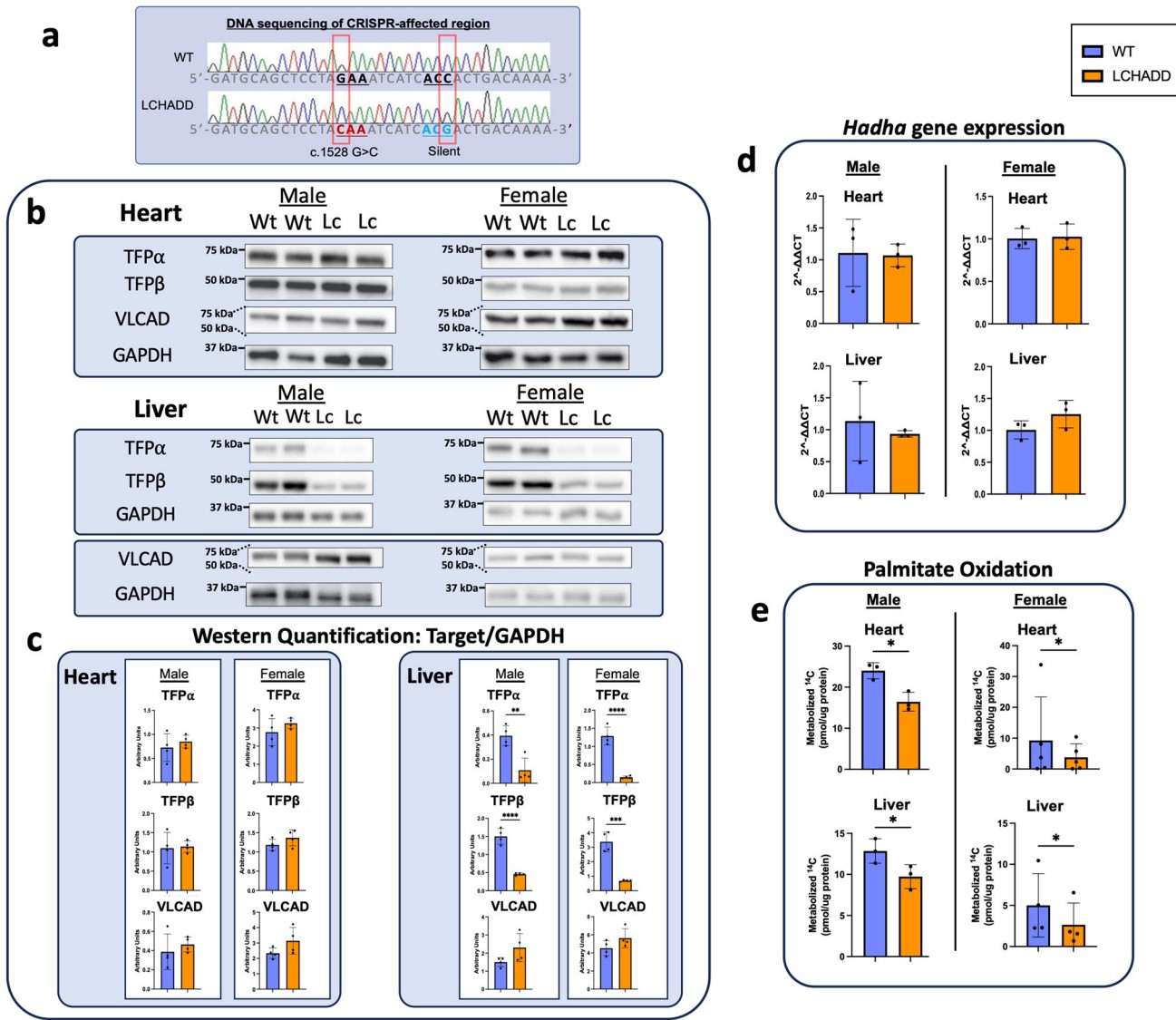

**Fig. 1 Biochemical analysis of LCHADD mutant mouse.** The G1528C mutation and a silent mutation were introduced to C57Bl/J6 mice by CRISPR-Cas9 to model human LCHADD. **a** Sanger sequencing of a small region of exon 15 of *Hadha* in WT or G1528C mice. Red boxes represent CRISPR/Cas9 edited nucleotides. **b** Cropped images and (**c**) quantification of Western blots for TFPα, TFPβ, and VLCAD protein from 15-month-old mice (Wt=wildtype; Lc=LCHAD). TFPα and TFPβ show decreased protein expression in the LCHADD liver but not heart in both sexes ($n = 4$ per group). Original blots can be found in Supplementary Fig. 6. **d** RT- qPCR of *Hadha* mRNA from 12–18-month-old mice show no change in gene expression ($n = 3$ per group). **e** Palmitate oxidation was measured by incubating radiolabeled palmitate with dounced tissue using age/sex matched pairs of each genotype. Heart and liver tissue metabolized-$^{14}$C are decreased in LCHADD mice (Male: age: 3–15 months; heart, liver $n = 3$ per group; Female: age: 3–6 months; heart $n = 5$, liver $n = 4$). Data are presented as mean + SD. Genotypes were compared by unpaired t-test by sex; *$p < 0.05$; **$p < 0.01$; ***$p < 0.001$; ****$p < 0.0001$.

Columbus Oxymax chamber and the data was analyzed in CalR software[14]. Male LCHADD mice had a higher respiratory exchange ratio (RER; $VCO_2/VO_2$) than WT mice during the dark period when mice are most active, suggesting increased glucose metabolism and decreased fat oxidation (Fig. 2a, b). This was due to higher $VCO_2$ during dark hours while the $VO_2$ was not different (Supplementary Fig. 3). RER was not different between female LCHADD and WT mice (Fig. 2c, d and Supplementary Fig. 3).

Patients with LCHADD accumulate partially oxidized long-chain 3-hydroxy-acylcarnitines in blood that is a biochemical marker for LCHADD. We measured plasma 3-hydroxy-acylcarnitines in blood from fasted LCHADD and WT mice by LC-MS/MS and found significantly higher long-chain 3-hydroxy-acylcarnitine species in blood from LCHADD male (Fig. 2e) and female mice (Fig. 2f); similar to levels observed in human patients

(C16:1OH mean = 0.13 and C18:1OH mean = 0.86 µmol/L)[15]. Higher RER and elevated long-chain 3-hydroxy-acylcarnitines suggest decreased whole body FAO with the accumulation of partially oxidized fatty acids similar to humans with LCHADD.

**LCHADD mutant mice exhibit lower ketone concentrations with fasting and impaired exercise tolerance.** Other mouse models of FAO disorders develop fasting-induced low blood glucose, and low ketones with or without the additional stress of a cold challenge[16]. We measured blood glucose and ketone concentrations in LCHADD and WT mice during an 18-h fast at room temperature. Serum glucose decreased with fasting in both groups over time (Fig. 3a). The 2-way ANOVA indicated an overall genotype effect suggesting LCHADD male and female mice had slightly lower serum glucose compared to WT male and

**Table 1 Pup viability by genotype and breeding strategy.**

| Pup Genotype | Het x Het | | Het x LCHADD | |
|---|---|---|---|---|
| | expected no. (ratio) | observed no. (ratio) | expected no. (ratio) | observed no. (ratio) |
| +/+ (WT) | 162 (25%) | 179 (30.8%) | – | – |
| +/− (Het) | 324 (50%) | 324 (55.8%) | 69 (50%) | 69 (73.40%) |
| −/− (LCHADD) | 162 (25%) | 78* (13.6%) | 69 (50%) | 25* (26.6%) |

Genotype numbers for het-het and het-LCHADD breeding strategies. Expected number based on Mendelian autosomal inheritance. Observed number given as total number and ratio of total pups. Stats by chi squared goodness of fit test ($\chi 2$).
*no.* number, *het* heterozygous, *WT* wildtype, *LCHADD* Long-chain 3-hydroxyacyl CoA dehydrogenase deficiency.
*$p < 0.001$.

female mice but post-hoc comparisons only observed a significant difference between LCHADD and WT females after 6 h of fasting (Fig. 3a). Total blood ketone concentrations increased with fasting in both groups over time but ketones were significantly lower in LCHADD male and female mice compared to WT mice (Fig. 3b). Overall, LCHADD mice exhibited hypoketosis with fasting compared to WT mice.

Previous FAO disorder mouse models also exhibited reduced exercise capacity during a moderate intensity exercise test and lower $VO_2$ max during maximal exercise[17]. Time to exhaustion with moderate intensity exercise was assessed and LCHADD mice exhausted much earlier than WT mice (Fig. 3c). During $VO_2$ max testing, LCHADD mice reached or neared lactate threshold, as indicated by a respiratory exchange ratio (RER) of >1.0, at 11 meters/min while WT mice never attained an RER above 0.9 (Fig. 3d). During the same $VO_2$ max exercise test, LCHADD mice achieved maximal $VO_2$ consumption at 5 to 7 meters/min that then decreased at higher treadmill speeds (Fig. 3e); maximal $VO_2$ was significantly lower in LCHADD mice in comparison to WT mice at 9 and 11 meters/min. This data indicates LCHADD mice experience decreased $VO_2$ max and impaired exercise tolerance as compared to WT littermates.

**Evidence of cardiomyopathy in LCHADD mutant mice**. Because dilated cardiomyopathy with systolic dysfunction is a common phenotype in human LCHADD patients, heart function was assessed. Two-dimensional and Doppler echocardiography was performed on 12 to 15 month old mice to assess left ventricular (LV) dimensions and function. Male and female LCHADD mice had lower ejection fraction associated with lower cardiac output and stroke volume (Fig. 4a–c), and increased left ventricular wall mass (Fig. 4d) compared to male and female WT mice. To support this finding, we observed an increase in heart tissue weight (Fig. 4e) and heart-to-body weight ratio (Fig. 4f) among LCHADD males and females compared to WT males and females. This suggests eccentric hypertrophy with a dilated cardiomyopathy phenotype. The cardiac dilation in our LCHADD mouse model may be related to our observed impaired exercise tolerance. Similar cardiac phenotypes have been observed in other related mouse models of FAO disorders[18,19].

**Neurological assessment of LCHADD mutant mice**. Because peripheral neuropathy is a phenotype in human LCHADD patients, a panel of neurological function tests was performed on 13-month-old mice. LCHADD male and female mice showed decreased wire hang fall and reach scores (Fig. 5a, b), indicating impaired motor function. LCHADD male and female mice also showed a non-significant trend for lower activity levels (Fig. 5c) and decreased time spent (Fig. 5d) in the center in an open field study, suggesting increased anxiety levels. In contrast to the wire

hang test, there was no genotype difference in rotarod performance or grip strength (Supplementary Fig. 4). Decreased motor function may also contribute to impaired exercise tolerance in LCHADD mice. Conversely, early exhaustion may contribute to changes observed in wire hang tests.

**Retinal phenotype of LCHADD mutant mice**. Because LCHADD patients develop a unique chorioretinopathy, we assessed retinal function in our LCHADD mouse. Visual parameters were tested in LCHADD and WT littermates of mixed sex at 1 year of age. Upon evaluating visual performance with optokinetic tracking (OKT) and retinal function with electroretinograms (ERG), LCHADD mice had significantly lower spatial frequency thresholds (Fig. 6a) and photopic b-wave amplitudes (52% reduction; Fig. 6b). As these two tests were evaluated in light-adapted conditions, these reduced responses indicate that cone mediated function is disrupted. Scotopic b-wave amplitudes, used to assess rod-mediated function, were not different (Supplementary Fig. 5a). However, scotopic c-waves, specifically measuring retinal pigment epithelium (RPE) function, were significantly reduced by 32% in LCHADD mice compared to WT mice (Fig. 6c) indicating RPE dysfunction. RPE degeneration was observed on fundus images as white or hypopigmented spots. When grading fundus images for presence or absence of these areas, 53% of LCHADD eyes evaluated had hypopigmented areas present. There were no hypopigmented areas observed in WT eyes (Fig. 6d, arrowheads). H&E staining of retinal cross sections revealed numerous large areas of RPE disruption in LCHADD mice (Fig. 6e, red arrows). H&E cross sections were scored based on the percentage of RPE containing these areas of RPE loss or disruption (range from 0–100%). Some areas were observed in WT mice but LCHADD had a significantly higher percentage of RPE containing visibly disrupted areas (80% in LCHADD compared to 40% in WT) suggesting greater RPE disruption (Fig. 6e). The nature of the RPE disruption is currently unknown but does not appear to be lipid deposits based on oil-red-O staining of retinal cross-sections. The RPE loss may be indicative of immune infiltration, or dead cells as previously reported in human eyes[20]. There was no difference in the RPE or retinal thickness on spectral domain-optical coherence tomography (SD-OCT) images suggesting normal retinal structure (Supplementary Fig. 5b). Overall, LCHADD mice display visual performance decline and retinal dysfunction with specific evidence of RPE damage.

**Discussion**

LCHADD is a severe long-chain FAO disorder characterized by hypoketotic hypoglycemia, dilated cardiomyopathy, exercise intolerance and recurrent rhabdomyolysis similar to other long-chain FAO disorders[5,21]. Patients with LCHADD also present with a unique retinopathy and peripheral neuropathy not apparent in the other FAO disorders[6,22]. Due to a lack of a LCHADD mouse model, we created a mouse model harboring the pathogenic variant c.1528 G > C using CRISPR/Cas9 technology. Our murine model recapitulates many of these features of the human disorder including LCHADD-specific phenotypes. The LCHADD mouse exhibited reduced whole body and tissue FAO; accumulates partially oxidized 3-hydroxy-acylcarnitines in blood similar to human patients; develops lower ketone concentrations with fasting; and exhibits exercise intolerance. Importantly, the LCHADD mouse has a retinal phenotype that has not been reported in other mouse models and thus represents a unique animal model of LCHADD retinopathy. LCHADD mice also demonstrate evidence of a dilated cardiomyopathy. Cardiac complications are the most common cause of death among

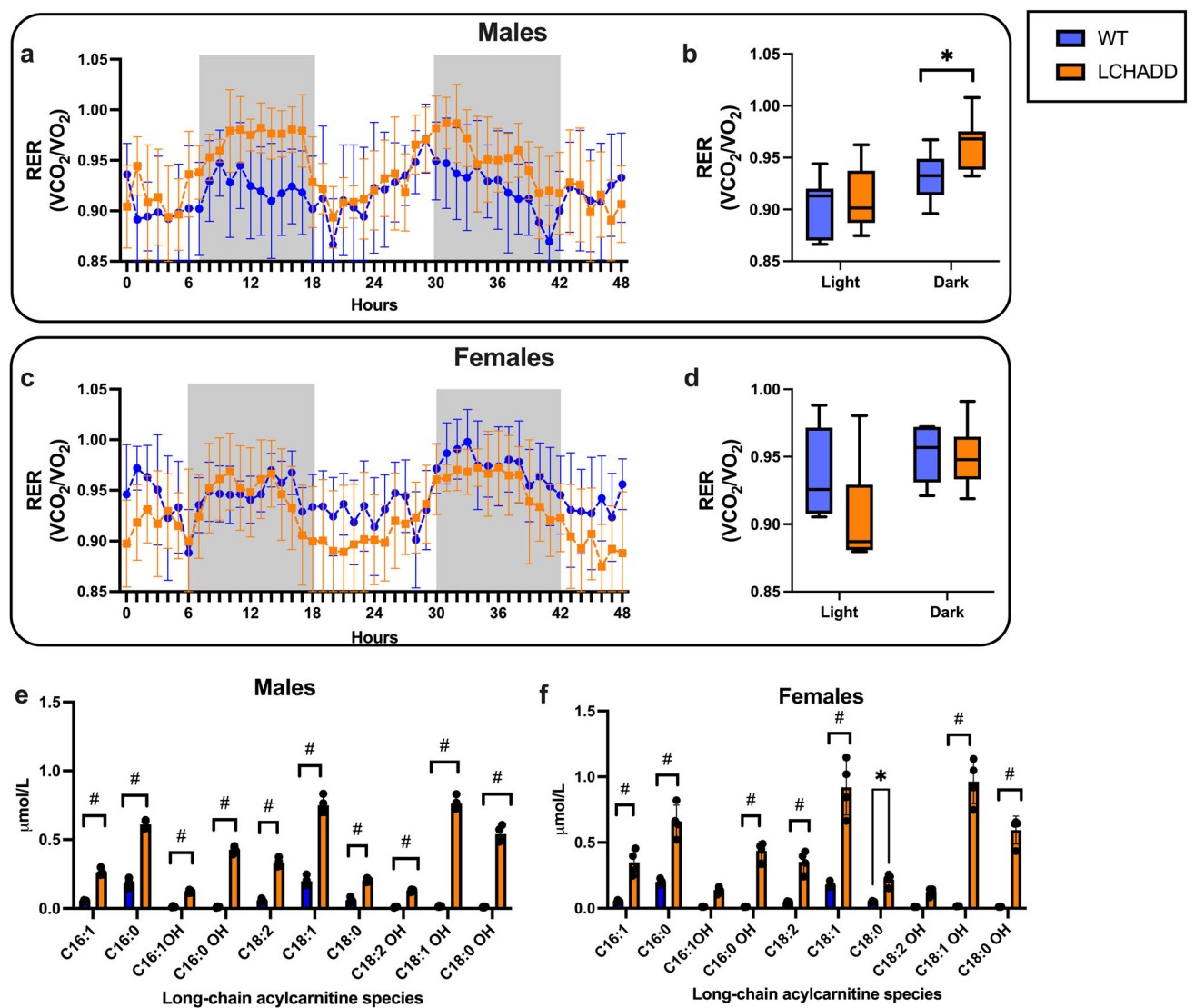

**Fig. 2 Resting substrate oxidation and serum acylcarnitines.** Ten-month-old LCHADD and WT mice were placed in Oxymax chambers for 4 days; data for last 48 h is displayed. Dark periods are indicated by gray shading regions. **a** Hourly Respiratory Exchange Ratio (RER; VCO$_2$/VO$_2$) and (**b**) light and dark period RER averages illustrate RER was higher in LCHADD males ($n = 7$) compared to WT males ($n = 7$) during the active/dark period. **c** Hourly RER and (**d**) light and dark period RER averages of LCHADD females ($n = 6$) were not different than WT females ($n = 6$). **e** LCHADD male mice ($n = 4$) and (**f**) LCHADD female mice ($n = 4$) had higher serum long-chain 3-hydroxyacylcarnitines compared to WT male ($n = 4$) and female ($n = 4$) mice. Data presented as mean ± SD. **a**, **c** Genotypes compared by repeated measures. **b**, **d** Box plots illustrates interquartile range; center line is median; whiskers are min and max. Genotypes compared by *t*-test. **e**, **f** Acylcarnitines compared by 2-way ANOVA; (main effects = genotype, species) with Sidak's multiple comparisons by sex; *$p < 0.05$; #$p < 0.0001$.

patients with LCHADD or TFPD. This mouse model could provide insight into cardiac pathophysiology when FAO is impaired and can be used as a model to test therapeutic interventions. LCHADD-associated peripheral neuropathy has a large impact on quality of life; a model to study this debilitating complication has been elusive. It is unclear at this point in time if the impaired motor function as assessed in the wire hang test and reduced activity levels in open field recapitulates features of the human disease. However, if these behavioral phenotypes prove to model the human disease complication, it opens opportunities to study and therapeutically target under-investigated aspects of this rare disease.

Mouse models have been crucial for studying long-chain FAO disorders; however, development of a mouse model that has a substantial decrease in FAO but is not neonatal lethal has been difficult[9,10,23,24]. The most popular mouse models for long-chain

FAO disorders are the very long-chain acyl-CoA dehydrogenase (VLCAD) and the long-chain acyl-CoA dehydrogenase (LCAD) knockout (KO) mice[11,12,25]. VLCAD and LCAD recapitulate some aspects of the human VLCAD deficiency; however, these models do not present with a phenotype as severe as the human deficiency potentially because VLCAD and LCAD appear to have overlapping function in mice while this overlap is not observed in humans[26]. This allows knockout mice to compensate, at least partially, for any loss of VLCAD or LCAD activity. Consequently, FAO is less impaired[16]. The LCHADD mouse model has a comparable, but in several ways more severe phenotype than VLCAD or LCAD KO mouse models potentially because there is no known protein that can compensate for the partial loss of LCHAD activity. LCHADD mice exhibit decreased FAO capacity as demonstrated by a higher RER in the LCHADD male mice, a phenomenon that has not been conclusively

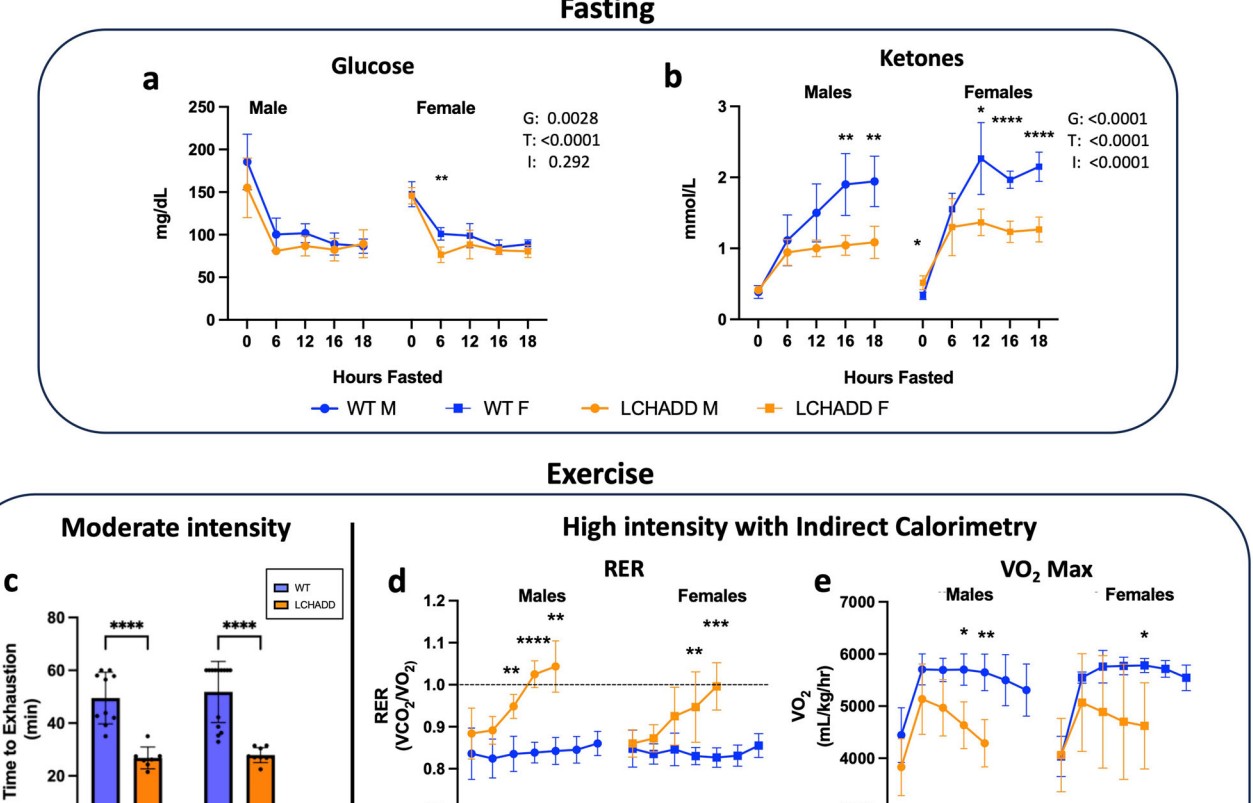

**Fig. 3 Fasting and exercise response.** LCHADD (males, $n = 7$; females, $n = 6$) and WT (males, $n = 7$; females, $n = 6$) mice were fasted overnight for 18-h. (**a**) LCHADD females had lower glucose at 6 h and (**b**) both sexes had lower ketones over time. (Main effects: G = group; T = time; I = interaction). **c** In a moderate intensity exercise test, mice were run until exhaustion or up to 60 min. LCHADD male ($n = 7$) and female ($n = 9$) mice exhausted earlier than WT male ($n = 10$) and female ($n = 14$) mice. A maximal exercise protocol was implemented until animals completed the protocol, exhausted, or reached a Respiratory Exchange Ratio (RER or $VCO_2/VO_2$) > 1.0. **d** LCHADD mice (males, $n = 5$; females, $n = 4$) exhibited consistently higher RER than WT mice (males, $n = 5$; females, $n = 4$). LCHADD males exceeded RER > 1.0, while some females exhausted before achieving RER of 1.0 (**e**) LCHAD mice also exhibit lower $VO_2$ max compared to WT. Data are presented as mean + SD. Genotypes were compared by repeated measures ANOVA (**a**, **b**, **d**, **e**) or 2-way ANOVA (**c**) with Sidak's multiple comparisons; *$p < 0.05$; **$p < 0.01$; ***$p < 0.001$; ****$p < 0.0001$.

observed in either VLCAD or LCAD KO mice[27–29]. Similarly, fasting-induced hypoketotic hypoglycemia is a common phenotype tested in murine FAO models. LCAD KO mice become hypoketotic and hypoglycemic after an overnight fast but VLCAD KO mice require both fasting and cold stress to induce hypoglycemia[27,30–32]. This LCHADD mouse, like the LCAD KO mouse, has lower blood ketones than WT mice with only an overnight fast and without any cold challenge, suggesting that there is a similar reliance on glucose with decreased ketosis during fasting when compared to the other models. The LCHADD mouse also experiences exercise intolerance like the VLCAD KO mouse[33,34]. Both LCHADD and VLCAD KO mice have a reduced ability to exercise for prolonged periods, and reach $VO_2$ max much earlier during exercise than WT[28,35]. All together, this mouse model exhibits multiple relevant disease specific manifestations of LCHADD.

We anticipated normal protein expression of TFPα in tissues that rely on FAO for energy similar to what has been reported in cultured human LCHADD fibroblasts[1,36]. It was thus surprising when we measured decreased TFPα/ TFPβ levels in the liver of LCHADD mice. It is unknown if TFPα expression in human tissues differ in LCHADD patients but it is possible that human patients with LCHADD may have lower liver expression of TFPα

and TFPβ compared to normal liver. Alternatively, it may also be unique to the mouse or related to the timing of tissue collection such as the impact of feeding/fasting on protein expression. Pulse chase experiments indicate the TFP complex has a relatively long half-life in fibroblasts, around 48 h[4,37,38], so the fed or fasted state may have no impact on protein levels but this has not been tested. Alternatively, the lower protein level with similar mRNA by qPCR could suggest decreased protein stability in the liver. Liver protein instability, but not heart, is a surprising finding that will be further investigated.

Chorioretinopathy is a unique complication of LCHADD not observed in other FAO disorders[20]. Our LCHADD mouse model exhibits a retinal phenotype with evidence of decreased visual performance, reduced retinal function (RPE and cones) and disruption of the RPE layer on H&E staining[8]. We have yet to elucidate the pathological etiologies of the LCHADD retinopathy in this mouse model. Two main hypotheses include (1) decreased FAO in the RPE disrupts the normal retinal metabolic ecosystem between RPE and photoreceptors and (2) selective retinal toxicity by LCHADD-specific circulating metabolites such as plasma long-chain 3-hydroxy fatty acids and/or 3-hydroxy-acylcarnitines[39]. Higher cumulative exposure to plasma 3-hydroxy-acylcarnitines is associated with decreased retinal

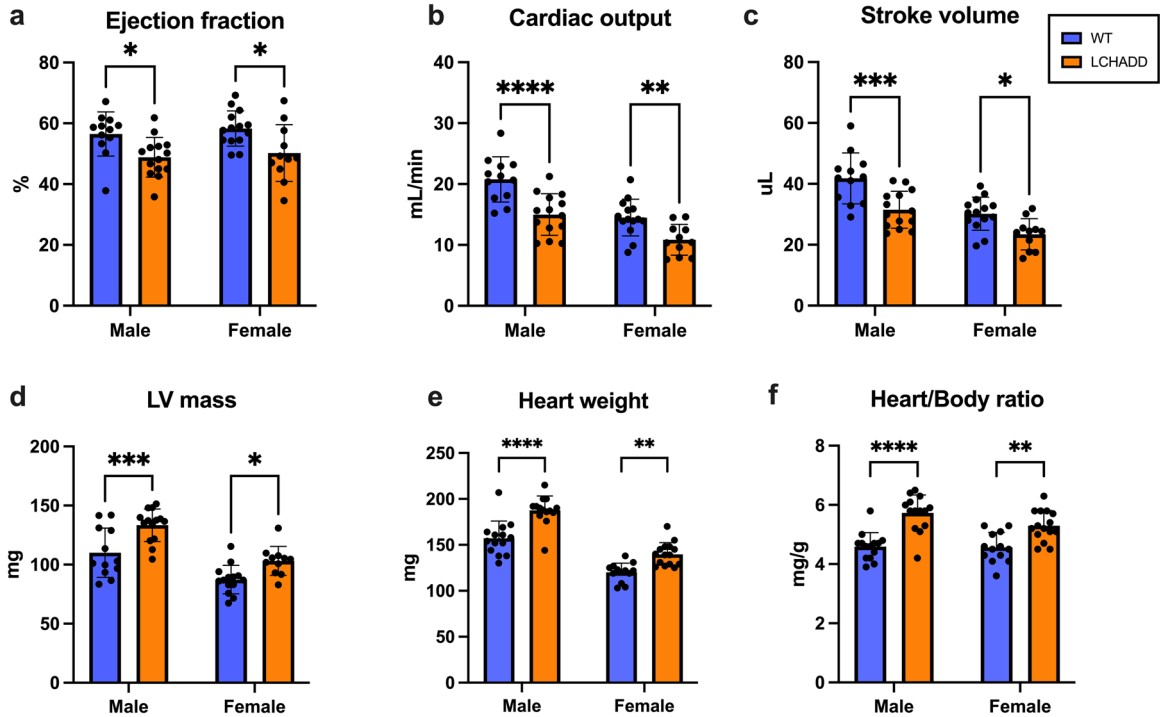

**Fig. 4 Cardiomyopathy observed in LCHADD mice.** Echocardiograms and tissue dissections ($n = 11$–15 per group) were completed on 12–15-month-old mice. **a** Ejection fraction, (**b**) Cardiac output and (**c**) Stroke volumes were lower in LCHADD males ($n = 14$) and females ($n = 11$) compared to WT (males $n = 12$; females $n = 14$). **d** Left Ventricular (LV) mass, (**e**) heart weight and (**f**) heart/body weight ratio were higher in LCHADD males and females compared to WT. Data are mean ± SD. 2-way ANOVA (main effects: sex, genotype) with post-hoc Sidak's multiple comparisons. ****$p < 0.0001$; ***$p < 0.001$; **$p < 0.01$; *$p < 0.05$.

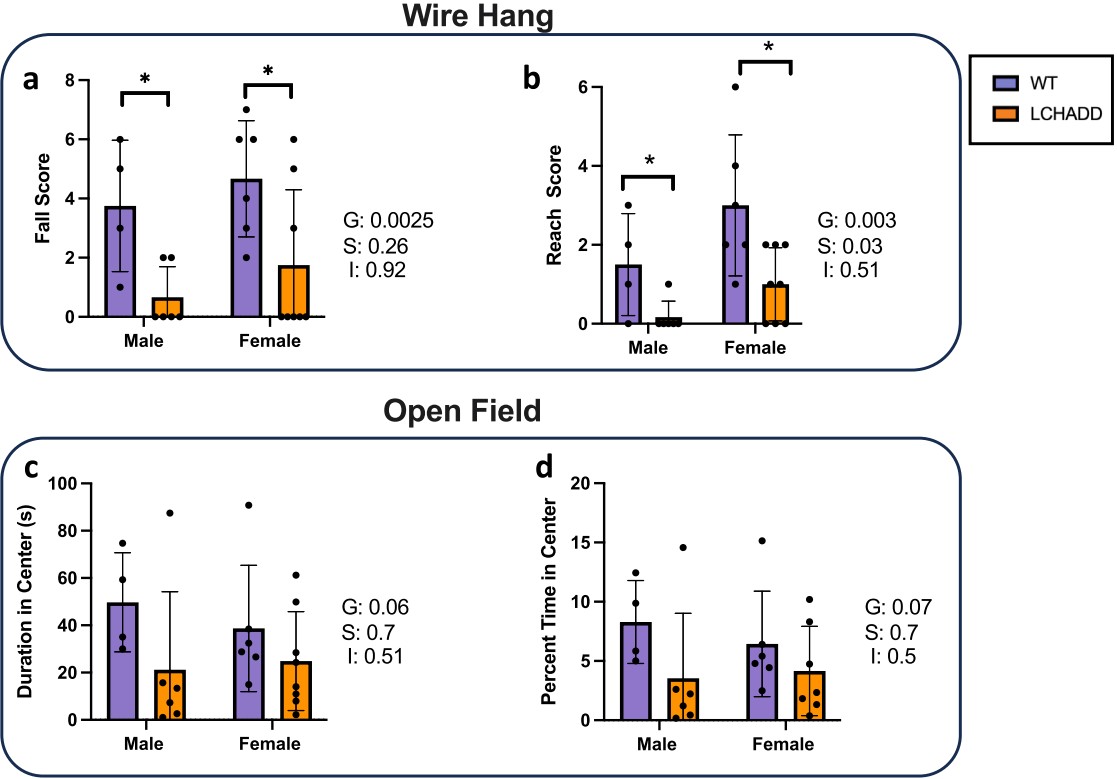

**Fig. 5 Neurological and behavioral testing.** Tests were performed in 1-year-old mice. (WT males $n = 4$, females $n = 6$; LCHADD males $n = 6$, female $n = 8$). **a, b** Wire hang tests showed a lower (a) fall score and (b) reach time in male and female LCHADD mice compared to WT male and female mice. Open field tests showed non-significant trends for decreased (**c**) duration in the center and (**d**) percent time in center in LCHADD mice compared to WT (2-way ANOVA, Main effects: G = genotype; S = sex; I = interaction). Data are mean ± SD. *$p < 0.05$.

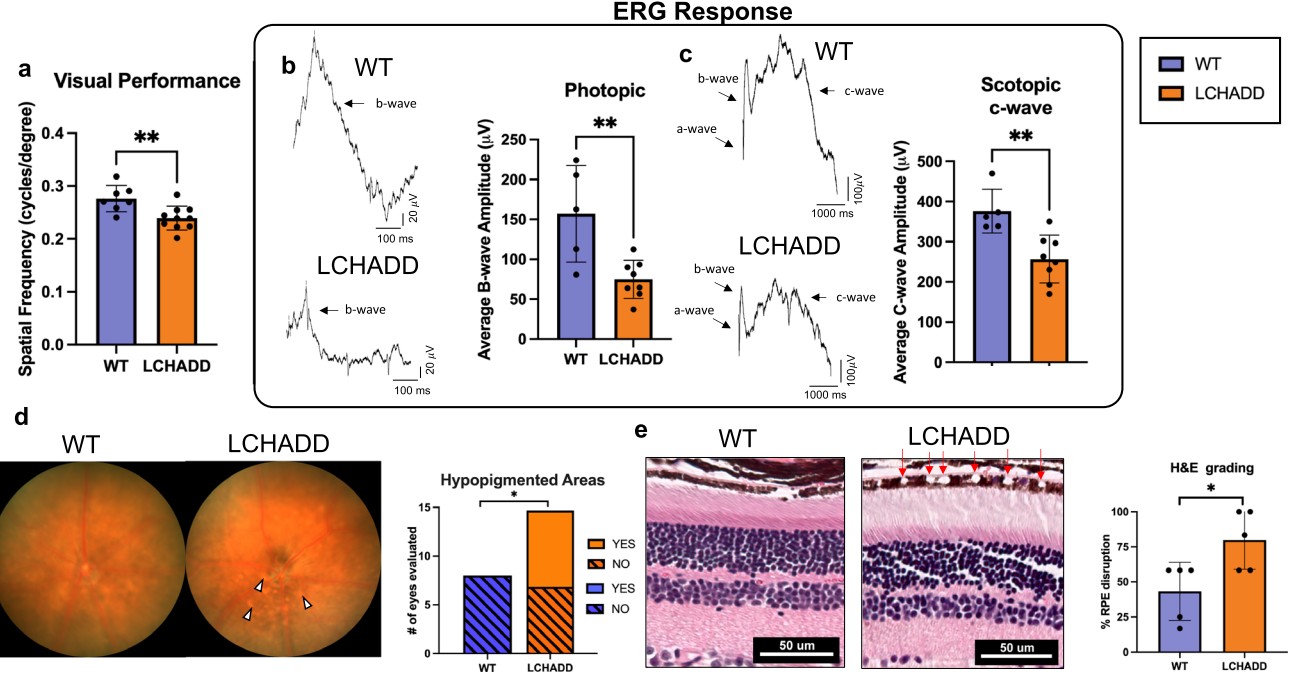

**Fig. 6 Visual parameters in 1 year old WT and LCHADD mice. a** Spatial frequency of LCHADD mice ($n = 10$) was lower than WT mice ($n = 7$). **b, c** (WT: $n = 5$, LCHADD, $n = 8$) (**b**) Wave forms and bar graph illustrate lower photopic b-wave amplitudes and (**c**) lower scotopic c-wave amplitudes at the highest light intensity evaluated. **d** Representative fundus images demonstrate hypopigmentation in LCHADD retina (arrowheads) indicative of retinal atrophy. Quantification of hypopigmented areas observed in LCHADD only. Yes = observed hypopigmented areas; No = no observed hypopigmentation. **e** H&E staining of retinal sections highlights large areas of RPE disruption (red arrows) that are visible in the RPE of LCHADD mice. % of RPE with areas of disruption was greater in LCHADD mice. Data are presented as count or as the mean ± SD. Compared by unpaired $t$-test or fishers exact test *$p < 0.05$; **$p < 0.01$.

function in humans[8,40]. However, these metabolites may simply be a biochemical marker for interrupted FAO in the body, and by extension, the retina. We are currently conducting a large natural history study of LCHADD retinopathy in humans and testing a retina-directed gene therapy treatment approach in our mouse model to further our understanding of LCHADD-associated retinopathy. In particular, we plan to use OCTA in future experiments to explore the choroidal phenotype in the LCHADD mice.

Cardiomyocytes preferentially oxidize long-chain fatty acids to generate the energy required for normal cardiac function[41], and patients with long-chain FAO disorders can present with heart failure, dilated cardiomyopathy, and arrhythmias during infancy, adolescence or later in life[42–44]. Symptomatic or asymptomatic ventricular dysfunction, from dilated or hypertrophic cardiomyopathy, is often the first manifestation among infants and children with LCHADD or TFPD[45,46]. Cardiac complications are the main cause of early demise in patients. The underlying etiology of cardiac complications in long-chain FAO disorders is hypothesized to be from decreased energy production in the heart or the accumulation of potentially toxic lipid intermediates[47]. Our LCHADD mouse develops dilated cardiomyopathy with eccentric hypertrophy and provides an in vivo model to investigate the cardiac complications associated with LCHADD.

Peripheral neuropathy impacts patient mobility and quality of life but little is known about the pathophysiology of this disease manifestation[6,7]. It appears in almost all infants around 18 months of age with a loss of peripheral deep tendon reflexes. Some patients never progress past this and the absent reflexes have little functional impact, but many patients exhibit a rapid progression during puberty with foot drop, numbness, and weakness that ultimately lead to reduced mobility[6,7]. Our mice do not develop immobility up to 1 year of age and the changes in neurological testing appear to be confined to performance in the wire hang and open field tests. The phenotypes appear to be more detectable in male mice and milder than observed in human patients. Additional studies, such as pathologic examination of nerve tissue, to determine if the LCHADD mouse neurological phenotype has some similarities to the human presentation are needed.

In conclusion, we generated an LCHADD mouse harboring the common G1528C mutation in exon 15 of *Hadha* using CRISPR/Cas9 technology. Mice homozygous for the LCHAD pathogenic variant recapitulate aspects of the human phenotype including decreased FAO, accumulation of plasma 3-hydroxy acylcarnitines, lower ketones with fasting, exercise intolerance, and cardiomyopathy. We present evidence of a neurological phenotype that needs further characterization. Additionally, these mice have a retinopathy that may be similar to that observed in human patients with LCHADD. We are currently using this mouse model to investigate tissue specific effects of decreased FAO compared to the effects of potentially toxic circulating metabolites such as 3-hydroxy acylcarnitines. Future studies using this unique model of LCHADD will further elucidate the underlying pathophysiology of LCHADD and can be used to evaluate various therapeutic strategies.

## Methods
All animal procedures were reviewed and approved by the OHSU IACUC (eIACUC #B11243). All experiments were performed in accordance with AAALAC and ARRIVE guidelines. Age, sex, and N of the mice for each experiment can be found in Supplementary Data 1.

**Mouse creation**. Mice were created through Cyagen Biosciences (Santa Clara, CA). The target c.1528G-C mutation and a silent mutation, which prevents recutting by sgRNA, was introduced into exon15 by homology-directed repair. Cas9 mRNA, sgRNA, and donor oligo were co-injected into C57Bl/6 J zygotes for knock-in mouse production. Injected zygotes were implanted in pseudo-pregnant females. Positive founders were bred to produce F1. All pups were genotyped by PCR/Sanger sequencing using primers 5'-CCAAACCACCCAAGCCTGACTCT-3', 5'-CAC-CACTACTGTCCGTTTTGGAGAC-3. Five potential off-target sites were analyzed and no additional mutations detected.

**Western blots**. ~20 mgs of snap-frozen tissue were homogenized in ice-cold RIPA lysis buffer (Thermo Scientific Cat #89900) using a stick homogenizer followed by 3 rounds of sonication (7 s on; ~1 min rest) at ~11 W. ~20–30 ng of protein per sample were electrophoresed using TGX Stain- Free gels (Bio-Rad) which allow visualization of total protein in the gel and on blot. After transfer to PVDF membrane, transferred total protein was visualized with a fluorescent dye (Bio-Rad) before blots were blocked in Tris buffered saline with 0.1% Tween (TBST) plus 5% milk and then incubated with appropriate primary (Polyclonal: HADHA, Thermo Scientific, PA527348; HADHB, Thermo Scientific, PA5117024; ACADVL, Invitrogen, PA529959. Monoclonal: HADHA (3E9B1), Proteintech, 50-173-6347; GAPDH, Santa Cruz Biotechnology, sc-365062 and sc-365062 HRP)) and appropriate secondary antibodies. Concentrations of the antibodies used in Western Blots are provided in Supplementary Table 1. Membranes used for successive blots were stripped using a 5%NaCl 5% Acetic acid solution. Target proteins were visualized using PicoPlus HRP Chemiluminescence kit (Thermo-Fisher). Size was estimated using Precision Plus Dual Color Standard (Bio-Rad).

Equipment and settings: All images are shown without additional contrast. Total protein was scanned on a GelDoc EZ Imager (Bio-Rad). Blots were scanned with an Azure Sapphire imager on chemiluminescent settings using wide range auto-exposure for bands and visible light settings for ladders. Image densitometry for bands and total protein was analyzed on Bio-Rad Image Lab 6.1. Ratio of HADHA, HADHB, and VLCAD densitometry normalized to GAPDH and to total protein were compared between genotypes by sex with similar results (Supplementary Data 1).

**RT-qPCR**. Total RNA from was extracted from ~20 mg of snap-frozen tissue using RNeasy Fibrous Tissue Mini Kit (Qiagen) with an on-column DNase treatment. RNA was converted to cDNA using the High Capacity RNA-to-cDNA Kit (Thermofisher). The cDNA was used to perform quantitative PCR on the QuantStudio 5 Real-Time PCR System (Thermofisher) with ~10 ng of cDNA, 250 nm of forward/reverse primers (Supplementary Table 2), and Power Syber Green PCR Master Mix (Thermofisher). Amplification was performed at 95 °C for 10 min, followed by 40 cycles of 95 °C for 15 s, 53 °C for 15 s, 60 °C for 45 s. Primers were designed to span across 2 exons of the gene (Supplementary Table 2).

**Metabolic Flux**. Long-chain fatty acid oxidation was measured as described before[48]. Mice were anesthetized with isofluorane and euthanized, and liver (200 mg) or heart (whole heart) were quickly extracted and homogenized using a Dounce homogenizer (5 strokes) in STE buffer (0.25 M sucrose, 10 mM Tris-HCL, 1 mM EDTA, pH 7.4). Homogenates were centrifuged at $450 \times g$ for 10 min at 4 °C. Samples (30 uL) from the homogenate supernatant were incubated with 370 uL of oxidation reaction mixture (100 mM sucrose, 10 mM Tris-HCl, 5 mM $KH_2PO_4$,

0.2 mM EDTA, 80 mM KCl, 1 mM $MgCl_2$, 2 mM L-Carnitine, 0.1 mM Malate, 0.05 mM Coenzyme A, 2 mM ATP, 1 mM DTT, 0.7% BSA/500 μM cold palmitate/0.4 μCi [1-$^{14}$C]palmitate, pH of 8) for 37 °C for 30 min. The reaction was stopped by adding samples to Eppendorf tube with 200 uL of 1 M perchloric acid and a Whatman filter-paper circle soaked with 20 uL of 1 M NaOH in the cap. After gentle shaking for one hour at room temperature, the Whatman filter-paper (containing released $CO_2$) was removed and the radioactivity was measured on a liquid scintillation counter. Eppendorf tubes were also centrifuged at $14,000 \times g$ for 10 min at 4 °C. 400 uL of supernatant (containing acid-soluble metabolites) was collected and the radioactivity was counted on a liquid scintillation counter. Total protein concentration measured from homogenate supernatant using the Bradford Assay (Bio-Rad) was used for normalization.

**Fasting**. 11-month old mice were fasted overnight from 6pm–12pm. Blood draws were taken every 6 h by pricking the tail vein or removing the clot using a ~19ga needle. Ketones and glucose were measured using a KetoMojo (https://keto-mojo.com Napa, California) device and appropriate micosampling strips. Similar to a handheld glucometer, KetoMojo uses microsamples of blood applied to a testing strip to measure concentrations of ketone or glucose enzymatically (beta-hydroxybutyrate dehydrogenase (HBDH) or Glucose Oxidase, respectively).

**Energy expenditure**. *Resting IC* Mice were placed in enclosed cages with food and water starting early afternoon. Oxygen consumption and $CO_2$ production was measured for 4 days (12-h light/dark cycle) by the Oxymax system (Columbus Instruments; Columbus OH). Initial 48 h of acclimation time was not used for calculations. Data was uploaded and analyzed in CalR, a web based analysis tool for indirect calorimetry data (www.CalRapp.org)[14].

**Moderate intensity treadmill**. Using an Exer 3/6 treadmill with electric stimulus (Columbus Instruments; Columbus OH) 3–4-month-old mice were trained for 3 rounds of ~12 min. Mice were then run starting at 4 m/min. Speed was increased by 2 m/min increments every 5 min until 16 m/min. Mice were additionally encouraged to run by light brushing on the hind quarters with paper towels. Mice were run for 60 min or until exhaustion, defined as an unwillingness to mount a run despite encouragement[17].

**High intensity treadmill**. 10–12 month old mice were trained on an enclosed, metabolic treadmill with electric stimulus (Columbus Instruments; Columbus OH), which takes indirect calorimetry measurements via the Oxymax system. Mice were run starting at 5 m/min and speed increased by 2 m/min every 3 min until 15 m/min or RER ≥ 1.0.

**Neurological testing**. For the behavioral studies, 10-month-old LCHADD (8 females and 6 males) and WT littermates (6 females and 4 males) were used. The mice were tested for performance in the wire hang test on 4/4/2021, for performance in the rotarod test on 4/6/21-4/8/21, and for activity in the open field and subsequently grip strength on 4/9/21, as described in detail below. All animal procedures were reviewed and approved by the OHSU IACUC and in accordance with AAALAC and ARRIVE guidelines. Researchers were blinded to the genotype throughout all experiments.

**Wire Hang**. Motor function was also assessed using the wire hang task, adopting the falls and reaches method described by van

Putten 2014[49] and the TREAT-NMD SOP DMD_M.2.1.004. It is summarized here. Mice were placed on a suspended metal wire so that they were hanging only by their front paws. In this method, mice start with a fall score of 10 and a reach score of 0. Over the duration of 180 s, mice lost 1 point from the score every time they fell and gained 1 point every time they reached one of the poles holding up the wire. The time of each fall or reach event was also recorded. Each time a mouse fell or reached, the timer was paused to replace the mouse in the center of the wire again. This test allows assessments of endurance and strength and exploring more complex motor coordination.

**Rotarod**. Sensorimotor function was assessed using the rotarod. Mice were placed on a rotating rod (diameter: 3 cm, elevated: 45 cm; Rotamex-5, Columbus Instruments, Columbus, OH, USA). Rotation speed started at 5.0 rpm and accelerated 1.0 rpm every 3 s. Latency to fall (s) was recorded using photo beams located below the rod. Mice received three trials each day for 3 subsequent days.

**Open field**. General locomotion and anxiety-like behavior was measured in an open environment (40.6 cm in length) with transparent walls. Animals were placed into the maze for a single 10-min trial. Movement of the mice was recorded using video tracking with Ethovision XT 7 software (Noldus Information Technologies, Wageningen, The Netherlands). Dependent measures recorded were distance moved and time spent in the more anxiety-provoking center of the open field.

**Grip strength**. We used a Harvard Apparatus grip test device for assessing grip strength in the mice. The grip strength meter allows the study of neuromuscular functions in rodents by determining the maximum force displayed by an animal. The grip strength meter was positioned horizontally and the mice held by the tail and lowered towards the apparatus. The mice were allowed to grasp the metal grid and pulled backward in the horizontal plane. The force applied to the grid just before the mouse lost its grip was recorded as the peak tension. We performed 3 consecutive measurements at one-minute intervals.

**Retinal evaluations**. *Optokinetic Tracking (OKT)* OKT thresholds were used to identify spatial frequencies of gratings (cycles/degree) which define visual performance for animals (OptoMotry; CerebralMechanics, Lethbridge, Alberta, Canada). Briefly, animals were placed on the pedestal in the OptoMotry system and given five minutes to acclimate to the new environment. A simple staircase method at 100% contrast in normal lighting conditions was used for testing. Right and left eyes were tested separately and averaged together to get one spatial frequency per animal.

**Electroretinography (ERG)**. Mice were dark-adapted overnight. Under dim red light, mice were anesthetized with ketamine (100 mg/kg) and xylazine (10 mg/kg). Bilateral platinum electrodes were placed on the corneal surface to record the light-induced retinal potentials. The reference and ground electrodes placed subcutaneously in the forehead and tail, respectively. Scotopic a and b-wave ERG responses were recorded at increasing light intensities from $-2.76$ to $3.39 \log cd \cdot s/m^2$. Scotopic c-wave flash responses were recorded with a separate flash at $1.51 \log cd \cdot s/m^2$. Animals were then light adapted with bright white light for 10 min and photopic flashes were recorded at increasing light intensities from $-0.23$ to $1.02 \log cd \cdot s/m^2$. Animals were recovered from anesthesia with an i.p. injection of atipamezole 1 mg/kg. Data presented in Fig. 6 includes the scotopic c-wave flash at $1.51 \log cd \cdot s/m^2$ and the maximum photopic flash at $1.02 \log cd \cdot s/m^2$.

Representative Scotopic a and b full waveforms are provided in Supplementary Fig. 5a.

**Fundus photography**. Live, in-vivo retinal imaging was performed with the Micron IV (Phoenix Research Laboratories, Pleasanton, CA). Mice were anesthetized with ketamine (100 mg/kg) and xylazine (10 mg/kg). Eyes were kept lubricated with 2.5% Hypromellose (Goniovisc). White light was used to acquire bright field images. Exposure settings were kept consistent between all animals. Images were scored yes/no for the presence of hypopigmented white spots by an independent reviewer.

**Histology**. Before enucleation, the superior edge of the eye was marked. Once enucleated, eyes were immediately placed in cold 4% paraformaldehyde and incubated for 2 h at 4 °C. Eyes were then placed in cassettes and stored in 70% ethanol at room temperature. Orientated eyes were processed and embedded in paraffin for sectioning (Tissue-Tek VIP 6, Tissue-Tek TEC 5; Sakura Finetek USA, Inc., Torrance, CA, USA). Sections were cut with a microtome to a thickness of 4 μm, stained with hematoxylin-eosin (H&E), and viewed on a Leica DMI3000 B microscope (Leica Microsystems GmbH, Wetzlar, Germany). All images were taken at a magnification of ×40. Presence of vacuoles in the RPE were scored by an independent reviewer. The percent of the RPE visible on H&E with the presence of vacuoles was calculated for each eye. The patches of with vacuoles may not be contiguous but were averaged across the whole RPE.

**Spectral domain optical coherence tomography (SD-OCT)**. Mice were sedated using 1.5% isoflurane delivered via a nose cone, corneas were anesthetized with 0.5% proparacaine, and pupils were dilated using a combination of 1% tropicamide and 2.5% phenylephrine. Artificial tears were used to maintain corneal clarity. Mice were seated in a Bioptigen AIM-RAS holder and SD-OCT images were obtained using an Envisu R2200-HR SD-OCT instrument (Bioptigen, Durham, NC).[50,51] Each eye was imaged using linear horizontal scans in the temporal and nasals quadrants and linear vertical scans in the superior and inferior quadrants. SD-OCT images were acquired when animals were 12 months old.

**Echocardiography**. High-frequency (40 MHz) two-dimensional and Doppler echocardiography was performed to assess the status of the aortic valve and LV (Vevo 2100, VisualSonics, Toronto, Canada). End-systolic and end-diastolic LV dimensions and wall thickness, and LV ejection fraction were measured from the parasternal long-axis view by the single plane modified Simpson's method. Stroke Volume was calculated as the product of the LV outflow tract cross-sectional area and time-velocity integral on pulsed-wave Doppler. Left ventricular mass was calculated by end-diastolic images in the mid-ventricular parasternal short-axis view by Equation 1:

$$1.05 \times (5/6A_1[l + t] - 5/6A_2[l])$$

where $A_1$ and $A_2$ are the cross-sectional areas for the epicardium and endocardium, respectively; $l$ is the distance from the apex to the mitral valve plane; and $t$ is mean wall thickness.

**Statistics and reproducibility**. Data were analyzed using SPSS (Chicago, IL), CalR (www.CalRapp.org) and Prism 9.0 software (GraphPad, San Diego, CA). Figures were generated using Prism software. Data were graphed as mean ± standard deviation of the mean with the individual data points. $p < 0.05$ was considered statistically significant.

Western protein densitometry normalized to GAPDH and RT-qPCR, were compared by unpaired *t*-test between genotypes-LCHADD and WT by sex. Palmitate oxidation was completed as matched pairs (genotype and sex) and statistically compared by paired *t*-test between genotypes by sex. Indirect calorimetry results were analyzed in CalR software. CalR uses a general linear model (GLM) with ANOVA for differences independent of mass such as RER and an ANCOVA for variables impacted by mass such as $VO_2$, $VCO_2$, and energy expenditure[14]. Plasma acylcarnitines were compared by 2-way ANOVA (main effect: genotype and acylcarnitine species) by sex with post-hoc Sidak multiple comparison test. Fasting glucose and ketones and high intensity exercise RER and $VO_2$ studies were compared by a repeated measures ANOVA. Main effect *p*-values for group (G; LCHADD or WT), time (T) and their interaction are reported with a post-hoc Sidak multiple comparison test indicating specific timepoint differences. Differences between LCHADD and WT mice by sex (male and female) were compared by a 2-way Analysis of Variance (ANOVA) with a post-hoc Sidak multiple comparison test for moderate intensity exercise, echocardiography outcomes, and tissue weights. Main effect p-values for group (G; LCHADD or WT), sex (S; male or female) and their interaction are reported. Neurological/behavior testing differences between LCHADD and WT mice by sex (male and female) were compared by a 2-way Analysis of Variance (ANOVA) with a post-hoc Least Squares Mean difference test. For behavior testing with changes over time such as rotarod testing, a 2-way repeated measures ANOVA compared group (G; LCHADD male, LCHADD female, WT male, WT female) over time (T) and their interaction with a post-hoc Least Squares Mean difference test was used. OKT, ERG, H&E disruption of RPE were compared by unpaired *t*-test between genotypes- LCHADD and WT. Presence or absence of hypopigmented areas were compared by Fishers exact test.

**Reporting summary**. Further information on research design is available in the Nature Portfolio Reporting Summary linked to this article.

## Data availability

All data generated or analyzed during this study are included in Supplementary Data 1.

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

## Acknowledgements

This study was funded by generous support from the Scully-Peterson foundation and the National Eye Institute (R01EY032889). Eye tissue processing and sectioning was supported by the Casey Eye Institute, Leonard Christenson Eye Pathology Laboratory. This work was supported by the National Institutes of Health (Bethesda, MD) P30 EY010572 core grant, the Malcolm M. Marquis, MD Endowed Fund for Innovation, and an unrestricted grant from Research to Prevent Blindness (New York, NY) to Casey Eye Institute, Oregon Health & Science University.

## Author contributions

G.G. helped with the design of mouse model, organized breeding of mice, conducted western blots, fasting and exercise experiments, worked on data analysis and assisted with writing the manuscript. S.B. conducted RT-qPCR and metabolic flux studies. G.E. assisted with mouse colony management, genotyping, data analysis and manuscript writing. T.D. assisted with genotyping, western blots, data analysis and manuscript writing. R.R. conducted visual acuity and retinal testing, and data analysis of the retinal phenotype. DW conducted visual testing and retinal testing. M.E.P. helped designed and coordinate retinal experiments, and data analysis. W.P. conducted echocardiography. J.R.L. analyzed echocardiography results and data analysis and assisted with manuscript writing. S.H. conducted the behavior testing. J.R. and S.H. analyzed the behavioral data and related data analyses. J.R. assisted with the manuscript writing. C.O.H. helped with the design of the mouse model, and overall study design and assisted with manuscript writing. M.B.G. designed the overall project, conducted data analysis and drafted the manuscript. All authors read, edited, and approved the manuscript.

## Competing interests

The authors declare no competing interests.
