## [Peer Review File · Communications Biology]

Reviewers' comments:

Reviewer #1 (Remarks to the Author):

This manuscript describes the first knock-in mouse model for long-chain 3-hydroxyacyl-CoA dehydrogenase

2 deficiency. Homozygous mice are born at lower than unexpected frequency, and the surviving mice have been investigated mainly for their clinical phenotypes. Many of the phenotypes resembling those described in patients have been identified in these mice, making them an interesting and useful model for future studies and therapy trials. Some confusion in interpretation brings the fact that some phenotypes are only identified in females and others only in males. All phenotypes are mentioned in abstract without gender distinction. It would be great if a figure summarizing which phenotypes were found in males and which in females was included, and also a figure which shows the age of mice at the time of different investigations.

Specific comments to address:

- Authors report that TFP α protein level is clearly reduced in mutant mouse liver, which is a very interesting finding, consistent in both genders. Quantification of other tested proteins should be included as it looks as if TFP β is also reduced in male liver. What was the age when these were analyzed?
- FAO measurements, males or females?
- Energy expenditure, it is unusual that results of mice of different ages are combined (7 months to 12 months). Is their weight taken into consideration? Overall the mouse weight is only mentioned as measured until the age of 14 weeks. Weight is needed for measuring energy expenditure. What could explain the gender difference here?
- Serum acylcarnities seem clearly altered as expected, although I would not include statistics for n=2. Is this males or females and what age? It is surprising that such a key result is presented with only a couple of samples, and would be great if could be increased.
- Fasting experiments: what is the age there?
- Fig, 3E and F, y-axis does not start from 0.
- Heart weight – what is the total body weight?
- Nerve conduction velocity measurement and/or nerve biopsy would be valuable in determining whether the mice have peripheral neuropathy

Reviewer #2 (Remarks to the Author):

In this paper, Gaston et al describe the generation and preliminary characterization of a new mouse model for long-chain-3-hydroxyacyl-CoA dehydrogenase (LCHAD) deficiency due to the common p.E510Q mutation. They convincingly demonstrate that this model recapitulates specific aspects of the human disease including neurological phenotypes and retinopathy. Therefore this is important work that sets the stage for studies that will increase our understanding of disease pathophysiology and for preclinical studies that evaluate novel treatment options. I have a few suggestions to further improve the content of the work:

Intro: "The common LCHAD pathogenic variant results in a glutamic acid to glutamine amino acid change (E510Q) in the LCHAD active site of TFP α reducing its activity but leaving the other two enzymatic activities relatively intact." This statement needs a reference.

Panel figure 1F is labeled as "LCHAD flux as percent of WT". It is not really LCHAD flux that is measured here, please reconsider.

The blood ketone measurement using KetoMojo. Does it measure total ketones or individual metabolites (3-hydroxybutyric, acetoacetic, other ketones)? I could not find any specific information on this tool so I would like if the authors provide a bit more information about this tool.

VO₂ in figure 3F was normalized to body weight. That is not considered optimal, but in this case I would be fine with it as long as the authors state that there was no difference in bodyweight between the wild-type and the mutant animals used for this analysis. If there is a significant difference in body weight another statistical analysis tool such as multiple regression analysis needs to be considered. The results in table 1 are reminiscent of the pup viability data of the LCAD KO model reported by Wood et al (PMID: 15308124) and confirmed by others. It be good to add a statement to that effect.

Reviewer #3 (Remarks to the Author):

Gaston et al create and characterize a mouse model of long-chain 3-hydroxyacyl-CoA dehydrogenase deficiency (LCHADD). LCHADD is caused by mutation in the HADHA gene that affects its dehydrogenase activity, but leaves its hydratase activity intact. The nature of the human mutation precludes use of knockout animals as models for the disease, and the authors correctly create a point mutation model. Unlike the knockouts which are lethal, the model presented in this manuscript recapitulates multiple aspects of the human disease. This makes it an excellent tool for studying the disease and testing therapies. Sex differences in some aspects of the phenotype also make the LCHADD mouse an interesting subject to study in regards of sex specific regulation of basic metabolic pathways. Overall this is important and interesting work. There are multiple deficiencies mainly related to the rigor with which the experiments were performed. Considering that the main focus is to introduce and describe a new model, it is important that the characterisation is done properly.

Major issues:

1. There are multiple issues/discrepancies with the western blots and the quantification of the protein levels (Figure 1 B and C) that require the experiments to be repeated and quantified again:

a) One problem is the use of total protein staining of a single membrane to normalize for loading of multiple membranes. Could the authors repeat the blots and use loading controls, like tubulin or actin, on each membrane and repeat the quantification?

b) There are serious issues with the barplot for the TFP-alpha levels in the female hearth. The western blot and quantification data in the supplement indicate five replicates for each group, wild type and LCHADD. Instead the wild type data shows six replicates and the mutant has four. A data point at ~500 arbitrary units appears to have been moved from the mutant to the wild type. This data point along with an apparent outlier at ~200 arbitrary units in the wild type samples (technical issue with loading for sample #3 in the wild type?) reduce the wild type average and increase the mutant average. This creates an impression that the protein levels are not different between wild type and mutant hearths that does not match the western blot images.

c) Several blots (for example the female hearth VLCAD blot on Supplementary figure 7) appear to be re-blots of other blots. Is that correct? If so, reblotting needs to be documented for all re-blots shown on the figures. Otherwise questions arise about the specificity of the antibodies and the varying patterns on the westerns.

d) The barplots for heart and liver need to be separated from each other as they represent quantification of different membranes and the data was not scaled to make them comparable.

2. The RT-qPCR data on figure 1 D is invalid due to issues with the primer design for the actin reference. The primers are perfect match for a region in the *Lrrc58* transcript (mouse genome assembly GRCm38 region chr16:37,882,922-37,883,022) in addition to the intended actin target (chr5:142,904,634-142,904,734). The primers need to be redesigned and RT-qPCR needs to be repeated. I would recommend redesigning the primers for both *HADHA* and actin, so that they are placed in separate exons. This will reduce potential interference from contaminating genomic DNA, as the intron will significantly increase the amplicon size. When designing primers for the actin gene, be aware of an alternative exon that is skipped in muscle tissue.

3. Figure 1 E claims to show palmitate flux, but the data does not contain a time component. Flux measures the amount of metabolite processed per unit of time. The data as shown represents steady state levels.

4. What was the sex of the animals used to produce the palmitate flux data on figure 1? Considering the sex differences observed in the subsequent figures this is an important issue.

5. There is only a 40% decrease in "palmitate flux" in the *LCHAD* samples compared to wild type but the protein level is reduced more than fivefold and the presence of mutation implies even further reduction in activity. How can such discrepancy be explained? Were there controls to ensure that CO_2 is effectively captured? For example, how much radioactivity was left in the acid insoluble fraction? Could you also discuss other beta-oxidation pathways that are unaffected by the mutation and may have contributed?

6. On figure 2 A and B the control animals do not show the expected diurnal cycling of the respiratory quotient that is due to ad libitum food consumption in the active cycle. Perhaps the mice were not adapted to the cage for a sufficiently long time or had trouble accessing the food.

7. The data on figure 2 C is derived from only two replicates. This is insufficient. It is also important to generate data from both males and females.

8. It is unclear why the 16 hour time points on Figure 3 A and B were extracted into separate bar charts. The data from supplementary figure 4 should be moved to figure 3A and the data from figure 3C should be moved to the plot on figure 3B.

9. The data on figure 3A is confusing. There is hypoglycemia in the mutant at early time points (6 hours), but not a late time point (16 hours, supplementary figure 4). If beta oxidation in the liver is reduced, then the hypoglycemia should be exacerbated at 16 hours as glucose levels will depend more on gluconeogenesis. Conversely, the hypoglycemia at 6 hours could be due to lack of glycogen stores or inability to mobilize them, but this is clearly not the case as the exercise data on Figure 6E shows that the animals can mobilize glycogen. How do the authors explain this conundrum?

10. Figure 3 E and F. Why were animals of different ages used for the two panels? As RER was measured on panel E, then the O_2 data should have been available for that age. Conversely, why show only the oxygen consumption for the one year old group? The authors should show the full data for both ages as it is very interesting if there are any age related changes in the mutants. In humans *LCHAD* is a disease that progresses with age.

11. The tests used for neurological assessment can be influenced also by faster exhaustion. Faster

exhaustion is an expected consequence of the defect in lipid metabolism the LCHADD animals have. At the same time the rotarod test did not show differences between genotypes. The authors should discuss this limitation, as it means that neuropathy has not been unambiguously confirmed in the mouse model.

12. Could the authors provide full ERG traces and A-wave measurements? The ERG traces should include marks indicating the position of the c-wave.

13. The authors speculate that voids in the RPE observed on the histology images are lipid deposits. This is an important aspect of the phenotype. Can they perform oil-red staining to determine if indeed there is lipid accumulation. In how many animals was the RPE defect observed?

14. Supplementary figure 5 is used as evidence that there are no differences in the OCT, but there are irregularities in the choroid of the LCHADD mice. Choroidal atrophy is a part of the LCHADD pathology in humans. Could the authors reexamine their histology slides to determine if the choroid is intact in the mutants?

Minor issues:

1. Line 56, could the authors provide a reference showing that the E510Q mutation reduces the TFP-alpha activity?

2. Supplementary figure 7B, the VLCAD blot for the liver appears to be shifted one lane to the right relative the line that demarcates the wild type and LCHADD samples.

3. On supplementary figures 7 A and B please indicate which parts of the blots were used to prepare figure 1.

4. Assuming TFP-alpha protein levels are reduced specifically in the liver but RNA expression and unchanged it would be interesting to test if the mutation does not result in tissue specific mis-splicing of the mutant exon. Could the authors perform RT-PCR with primers placed in the flanking exons to determine if the mutant exon is included at similar levels and appropriate size in both heart and muscle?

RER and RQ are both used in the manuscript but they indicate the same $V(\text{CO}_2)/V(\text{O}_2)$ ratio. The authors should chose one term to use in the manuscript.

5. Data on figure 4 shows consistent trends that do not reach statistical significance in some of the experiments. This seems to largely be due to high variance. The high variance can potentially be remedied with more replicates. It would be nice if the authors add more replicates to the experiments.

Response to Reviewers comments	
Referee #1	
1. Authors report that TFPa protein level is clearly reduced in mutant mouse liver, which is a very interesting finding, consistent in both genders. Quantification of other tested proteins should be included as it looks as if TFPb is also reduced in male liver.	Other tested proteins were quantified in our new western blots and are illustrated in the revised figure 1 B,C. Both TFPa and TFPb are reduced. Please see figure 1, line 81.
2. What was the age when these were analyzed?	Age of animals for each outcome are listed in the first tab of data transparency excel file. The mice were 15 months old when tissues were taken for western blots.
3. FAO measurements, males or females?	The original figure was a mix of sexes. We have increased the number of FAO measurements and now present the data separated by sex in figure 1E, line 81.
4. Energy expenditure, it is unusual that results of mice of different ages are combined (7 months to 12 months). Is their weight taken into consideration?	We repeated the IC experiment in a new cohort of mice, all the same age. The revised figure 2, line 164 contains data from LCHADD and WT mice age 10 months. Weight was not different between genotypes but is considered as a covariate in CalR analysis of the IC data.
5. Overall the mouse weight is only mentioned as measured until the age of 14 weeks. Weight is needed for measuring energy expenditure. What could explain the gender difference here?	Thank you for this point. It is unclear the exact reason for the difference between males and females. It is possible that whole body oxidation that is relatively lower in females is not able to detect subtle differences in substrate oxidation. It is also possible that the FAO system is not engaged as much in females due to small frequent eating events with low physical activity.
6. Serum acylcarnities seem clearly altered as expected, although I would not include statistics for n=2. Is this males or females and what age? It is surprising that such a key result is presented with only a couple of samples, and would be great if could be increased.	We analyzed fasted serum from n=4 male and female mice of each genotype for plasma acylcarnitines. The mice were 15 month old at the time blood was collected. Figure 2 has updated figures in 2E and 2F by sex, line 164.
7. Fasting experiments: what is the age there?	We repeated the fasting experiments in a new cohort of animals all age 11 months at the time of fasting. Mice were fasted for 18 hours and the data is shown in the revised figure 3 A, B, line 180.
8. Fig, 3E and F, y-axis does not start from 0	Figure 3 IC, line 180 data for the VO₂ max testing is zoomed in to illustrate genotype differences, which is typical of IC data reporting.
9. Heart weight – what is the total body weight?	Total body weight was lower in females compared to males but not different between genotypes. We

	have added heart weight/ body weight ratio to figure 4, line 227.
10.Nerve conduction velocity measurement and/or nerve biopsy would be valuable in determining whether the mice have peripheral neuropathy	Thank you for this suggestion. We are exploring potential collaborations to conduct these studies. We plan to address in future experiments.
Referee #2	
1.Intro:” The common LCHADD pathogenic variant results in a glutamic acid to glutamine amino acid change (E510Q) in the LCHAD active site of TFP α reducing its activity but leaving the other two enzymatic activities relatively intact.” This statement needs a reference.	Added the following reference to line 29. Ijlst, L., Ruiten, J. P., Hoovers, J. M., Jakobs, M. E. & Wanders, R. J. Common missense mutation G1528C in long-chain 3-hydroxyacyl-CoA dehydrogenase deficiency. Characterization and expression of the mutant protein, mutation analysis on genomic DNA and chromosomal localization of the mitochondrial trifunctional protein alpha subunit gene. J Clin Invest 98 , 1028-1033. (1996).
2.Panel figure 1F is labeled as “LCHADD flux as percent of WT”. It is not really LCHAD flux that is measured here, please reconsider.	Thank you for the comment. We changed the graph to state “palmitate oxidation”. It represents the amount of palmitate oxidized in 30 minutes. Please see revised figure 1, line 81.
3.The blood ketone measurement using KetoMojo. Does it measure total ketones or individual metabolites (3-hydroxybutyric, acetoacetic, other ketones)? I could not find any specific information on this tool so I would like if the authors provide a bit more information about this tool.	Keto-mojo is a handheld meter that measures b-hydroxybutyrate via an enzymatic reaction. Similar to a glucometer, it can measure ketones with a small drop of blood. https://keto-mojo.com https://keto-mojo.com/wp-content/uploads/2020/09/GK-Ketone-Strip-Insert-English.pdf We added more details to the methods. Line 455-460
4.VO ₂ in figure 3F was normalized to body weight. That is not considered optimal, but in this case I would be fine with it as long as the authors state that there was no difference in bodyweight between the wild-type and the mutant animals used for this analysis.	Thank you for the comment. We repeated the indirect calorimetry results and loaded the data into Calr for analysis. https://calrapp.org Calr looks at body weight in the ANCOVA model of IC data. In our experiment there was no difference in body weight between male LCHADD and WT mice or between female LCHADD and WT mice. This graph has been added as supplemental figure 3, line 180.
5.The results in table 1 are reminiscent of the pup viability data of the LCAD KO model reported by Wood et al (PMID: 15308124) and confirmed by others. It	Thank you for the suggestion. A statement was added. “Week old homozygous mouse pups were identified at ~50% lower frequency than expected regardless of breeding approach suggesting

would be good to add a statement to that effect.	reduced LCHADD mouse pup viability similar to observations in the long-chain acyl-CoA dehydrogenase (LCAD) knockout mouse (Table 1). ¹⁷ line 74
Referee #3	
1. There are multiple issues/discrepancies with the western blots and the quantification of the protein levels (Figure 1 B and C) that require the experiments to be repeated and quantified again.	Western blots were repeated. Please see revised figure 1 line 81 and supplemental figure 6.
a) One problem is the use of total protein staining of a single membrane to normalize for loading of multiple membranes. Could the authors repeat the blots and use loading controls, like tubulin or actin, on each membrane and repeat the quantification?	Thanks for the comment. To clarify, each membrane used for probing was first stained for total protein using a fluorescent dye from Biorad. Then the membrane was repeatedly re-probed with indicated antibodies. Normalization had taken place on the actual membrane used. However, we did repeat the westerns and used GAPDH as a loading control. When we normalize to either total protein or GAPDH on the blots, we get a similar result. Please see revised figure 1 and supplemental figure 6, and supplementary data file.
b) There are serious issues with the barplot for the TFP-alpha levels in the female hearth. The western blot and quantification data in the supplement indicate five replicates for each group, wild type and LCHADD. Instead the wild type data shows six replicates and the mutant has four. A data point at ~500 arbitrary units appears to have been moved from the mutant to the wild type. This data point along with an apparent outlier at ~200 arbitrary units in the wild type samples (technical issue with loading for sample #3 in the wild type?) reduce the wild type average and increase the mutant average. This creates an impression that the protein levels are not different between wild type and mutant hearts that does not match the western blot images.	Bar plots of all of the proteins were created with data from the new blots. Each protein densitometry quantification was normalized to GAPDH and illustrated on separate bar graphs and compared with a t-test between LCHADD and WT. There are separate blots for males and females. n=4 replicates for each sex/genotype were evaluated.
c) Several blots (for example the female hearth VLCAD blot on Supplementary	We tried to indicate this in the legend. Our language may not have been optimal. However, new Westerns were done in which we stripped

figure 7) appear to be re-blots of other blots. Is that correct? If so, reblotting needs to be documented for all re-blots shown on the figures. Otherwise questions arise about the specificity of the antibodies and the varying patterns on the westerns.	between probings which should clarify interpretation. However our polyclonal TFPa does find multiple non-specific bands or, at least, bands of unknown significance in mouse liver. To validate the liver TFPa data we also provided a separate blot using both sexes and genotypes using a monoclonal TFPa antibody. (Supplemental figure 6C)
d) The barplots for heart and liver need to be separated from each other as they represent quantification of different membranes and the data was not scaled to make them comparable.	Bar plots have been separated as suggested.
2. The RT-qPCR data on figure 1 D is invalid due to issues with the primer design for the actin reference. The primers are perfect match for a region in the Lrrc58 transcript (mouse genome assembly GRCm38 region chr16:37,882,922-37,883,022) in addition to the intended actin target (chr5:142,904,634-142,904,734). The primers need to be redesigned and RT-qPCR needs to be repeated. I would recommend redesigning the primers for both HADHA and actin, so that they are placed in separate exons. This will reduce potential interference from contaminating genomic DNA, as the intron will significantly increase the amplicon size. When designing primers for the actin gene, be aware of an alternative exon that is skipped in muscle tissue.	Thank you for the comment. RT-qPCR primers were redesigned as suggested. The new primers are listed in Table 2, line 437. New RT-PCR data using the revised primers is now provided in figure 1, line 81.
3. Figure 1 E claims to show palmitate flux, but the data does not contain a time component. Flux measures the amount of metabolite processed per unit of time. The data as shown represents steady state levels.	We changed the label to palmitate oxidation. The results represents metabolized ¹⁴C in 30 minutes incubation in fresh tissue homogenates.
4. What was the sex of the animals used to produce the palmitate flux data on figure 1? Considering the sex differences	We increased the number of samples per sex and now separate palmitate oxidation into male and female graphs.

observed in the subsequent figures this is an important issue.	
5. There is only a 40% decrease in “palmitate flux” in the LCHAD samples compared to wild type but the protein level is reduced more than fivefold and the presence of mutation implies even further reduction in activity. How can such discrepancy be explained? Were there controls to ensure that CO₂ is effectively captured? For example, how much radioactivity was left in the acid insoluble fraction? Could you also discuss other beta-oxidation pathways that are unaffected by the mutation and may have contributed?	This is an interesting point. We have assumed that palmitate oxidation is testing enzyme activity at V_{max} and that most enzymes operate at much lower total flux in vivo. There are publications to suggest that peroxisomal oxidation can compensate to some degree for low mitochondrial β-oxidation and this could contribute to some of the discrepancy. The graphs illustrate CO₂ capture and acid soluble products. The samples for WT and LCHADD were treated the same but other controls were not included in the assay. Each assay was conducted on genotype, age and sex matched paired samples. Text changes: “Both liver and cardiac tissue exhibited lower metabolized ¹⁴C compared to those seen in liver and cardiac tissue of WT mice (Figure 1E). Palmitate oxidation was approximately 60% of WT suggesting reduced but not absent FAO capacity. A 60% residual palmitate oxidation seems relatively high in LCHADD. However, our assay measures total metabolized ¹⁴C including oxidation from β-oxidation in the peroxisome and ω-oxidation in microsomes; peroxisomal oxidation may compensate for lower mitochondrial oxidation in tissues with impaired mitochondrial β-oxidation.” Line 121-126
6. On figure 2 A and B the control animals do not show the expected diurnal cycling of the respiratory quotient that is due to ad libitum food consumption in the active cycle. Perhaps the mice were not adapted to the cage for a sufficiently long time or had trouble accessing the food.	We repeated the indirect calorimetry experiment with a new cohort of animals, all the same age (10 months old) including WT males n=7; females n=6 and LCHADD males n=7 and females n=6. Animals were adapted to the cage for about 48 hours and data for an additional 48 hours is displayed. Output was analyzed in CalR and VO₂, VCO₂, energy expenditure and body weights are included in supplemental figure 3. We believe the expected diurnal cycle was masked when we combined the 2 different aged data sets in the previous version of the manuscript. This is much cleaner and clear data. The result is essentially the same, male LCHADD mice have higher RER in the dark but no differences were observed in females.
7. The data on figure 2 C is derived from only two replicates. This is insufficient. It is also important to generate data from both males and females.	We analyzed blood from n=4 mice of each genotype and sex. The new analysis is displayed in Figure 2E,F, line 164.
8. It is unclear why the 16 hour time points on Figure 3 A and B were	We removed the 16 hour time point. Fasting studies were repeated with a new cohort of mice.

extracted into separate barcharts. The data from supplementary figure 4 should be moved to figure 3A and the data from figure 3C should be moved to the plot on figure 3B.	Mice were fasted for 18 hours and the new data is illustrated in Figure 3A, B, line180
9. The data on figure 3A is confusing. There is hypoglycemia in the mutant at early time points (6 hours), but not a late time point (16 hours, supplementary figure 4). If beta oxidation in the liver is reduced, then the hypoglycemia should be exacerbated at 16 hours as glucose levels will depend more on gluconeogenesis. Conversely, the hypoglycemia at 6 hours could be due to lack of glycogen stores or inability to mobilize them, but this is clearly not the case as the exercise data on Figure 6E shows that the animals can mobilize glycogen. How do the authors explain this conundrum?	We can speculate about glycogen stores but we have not measured this in our mice. It is possible the early decrease in glucose at 6 hours represents initial reliance on glucose oxidation before counter regulatory hormones fully activate glycogenolysis/ gluconeogenesis. The rebound and stabilization of glucose most likely is related to these processes. Other studies in VLCAD and LCAD mice suggest normal glycogen stores when fed but more rapid utilization and depletion with fasting.
10. Figure 3 E and F. Why were animals of different ages used for the two panels? As RER was measured on panel E, then the O2 data should have been available for that age. Conversely, why show only the oxygen consumption for the one year old group? The authors should show the full data for both ages as it is very interesting if there are any age related changes in the mutants. In humans LCHADD is a disease that progresses with age.	Different animals were not used in the VO₂ testing; the data is from the same animals illustrating lactate threshold with RER and VO₂ max with oxygen consumption.
11. The tests used for neurological assessment can be influenced also by faster exhaustion. Faster exhaustion is an expected consequence of the defect in lipid metabolism the LCHADD animals have. At the same time the rotarod test did not show differences between genotypes. The authors should discuss	We added this to the neurological section. Line 223-225” Decreased motor function may also contribute to impaired exercise tolerance in LCHADD mice. Conversely, early exhaustion may contribute to changes observed in wire hang tests.”

this limitation, as it means that neuropathy has not been unambiguously confirmed in the mouse model.	
12. Could the authors provide full ERG traces and A-wave measurements? The ERG traces should include marks indicating the position of the c-wave.	Full ERG traces have been added and have been clearly marked for the a, b, and c waveforms Figure 6B, C. Scotopic a-wave and b-wave measurements were not different between WT and LCHADD animals and thus not reported, but can be appreciated by the ERG traces in supplemental figure 5. We only found a difference in photopic b-wave amplitudes and scotopic c-wave amplitudes, which are reported in Figure 6, line 278. Our photopic flashes do not generate a measurable a-wave, so we only report a photopic b-wave.
13. The authors speculate that voids in the RPE observed on the histology images are lipid deposits. This is an important aspect of the phenotype. Can they perform oil-red staining to determine if indeed there is lipid accumulation. In how many animals was the RPE defect observed?	We instituted a scoring system for hypopigmented area on fundus images. An independent observer scored images yes/no for the presence of hypopigmented areas (WT n=8; LCHADD n=15 images). 53% of LCHADD images had hypopigmented spots but none of the WT images. (Figure 6D) We also completed a scoring of H&E cross-section. The grading for H&E was based on the % of the RPE with visible RPE drop-out or disruption. Five images of each genotype were scored. Some disruption were noted in WT cross-sections but LCHADD cross-section scores were significantly higher suggesting more of the RPE contained the disruption illustrated in Figure 6. We have not been able to identify the voids in the RPE at this point. We tried oil Red-O, and Bodipy in fixed cross-sections, and Bodipy in flat-mounts. An extensive explanation of our investigation is included at the end of this table. Please see below. At the end of this investigation we must conclude the voids are not lipid based. We revised the sentence: The sentence was revised: "The nature of RPE disruption is currently unknown but does not appear to be lipid deposits based on oil-red-O staining of retinal cross-sections (data not shown). The RPE loss may be indicative of immune infiltration, or dead cells as previously reported in human eyes.³" Line 266-269
14. Supplementary figure 5 is used as evidence that there are no differences in the OCT, but there are irregularities in the choroid of the LCHADD mice.	Choroidal atrophy is indeed part of the human phenotype. To address this comment, we looked at the SD-OCT images and the H&E images, but we did not see any significant differences between WT and LCHADD with the methods used. OCTA would

Choroidal atrophy is a part of the LCHADD pathology in humans. Could the authors reexamine their histology slides to determine if the choroid is intact in the mutants?	be a superior method to answer this question, but we do not currently have the capability to perform OCTA imaging in mice. At this point in time, we cannot tell if the choroid is impacted in our LCHADD mutant mouse. Future studies will address this question using an OCTA system with 1050nm wavelength light.
15. Line 56, could the authors provide a reference showing that the E510Q mutation reduces the TFP-alpha activity?	Added the following reference to line 29. Ijlst, L., Ruiter, J. P., Hoovers, J. M., Jakobs, M. E. & Wanders, R. J. Common missense mutation G1528C in long-chain 3-hydroxyacyl-CoA dehydrogenase deficiency. Characterization and expression of the mutant protein, mutation analysis on genomic DNA and chromosomal localization of the mitochondrial trifunctional protein alpha subunit gene. J Clin Invest 98, 1028-1033. (1996).
16. Supplementary figure 7B, the VLCAD blot for the liver appears to be shifted one lane to the right relative the line that demarcates the wild type and LCHADD samples.	New supplemental images of the blots are now provided. Please see supplemental figure 6
17. On supplementary figures 7 A and B please indicate which parts of the blots were used to prepare figure 1.	Part of the blots in figure 1 are noted on the supplemental images. A red box on the blots in supplemental figure 6 indicates the image shown in figure 1.
18. Assuming TFP-alpha protein levels are reduced specifically in the liver but RNA expression and unchanged it would be interesting to test if the mutation does not result in tissue specific mis-splicing of the mutant exon. Could the authors perform RT-PCR with primers placed in the flanking exons to determine if the mutant exon is included at similar levels and appropriate size in both heart and muscle?	Thanks for the suggestion. We do not think there is exon skipping in the liver because we do see a protein at the same size as WT. Exon 15 is 141bp and if it was skipped, we would anticipate a protein about 47 amino acids and 5kDa smaller. We will explore this further in future experiments.
19. RER and RQ are both used in the manuscript but they indicate the same $V(\text{CO}_2)/V(\text{O}_2)$ ratio. The authors should choose one term to use in the manuscript	We changed the manuscript to use RER throughout. As a note of nomenclature, typical energy balance physiologists use RQ to indicate gas ratio measures at rest and exercise physiologists use RER to indicate the same ratio during exercise. To avoid confusion, we have selected RER in this study.
20. Data on figure 4 shows consistent trends that do not reach statistical significance in some of the experiments.	Thank you for the suggestion. We added another cohort of animals and differences in all parameters now reach significance. Numbers for each

This seems to largely be due to high variance. The high variance can potentially be remedied with more replicates. It would be nice if the authors add more replicates to the experiments.

genotype and sex are n= 11-14. Please see revised figure 4, line 227.

RPE H&E vacuole/RPE disruption identification: We have tried many different techniques but are still unable to definitively identify the vacuoles or RPE loss observed in the LCHADD RPE on H&E cross-sections. The descriptions below provide a narrative of our process.

1. We first tried staining retinal cross sections with Oil Red O (ORO). This was unable to identify the vacuoles because the pigmentation of the RPE layer obscured any potential ORO staining specifically in the RPE layer.

Figure 1: Images of ORO staining of retinal cross-sections from A) WT mice fed a normal diet, B) LCHADD mice fed a normal diet, and C) WT mice fed a high fat diet. White arrows indicate the RPE layer; red=Oil Red O staining

2. We then tried staining retinal cross sections with BODIPY 493/503, a fluorescent marker of neutral lipids. This stain did not work because the varying autofluorescence from different cross-sections. Also, the bright autofluorescence from the photoreceptor outer segments obscured signals coming from the RPE and complicated quantification of lipid droplets specifically in the RPE.

Figure 2: Images of BODIPY 493/503 staining (green fluorescence) of retinal cross-sections from (A) WT mice, (B) LCHADD mice, and (C and D) WT mice with no BODIPY staining (negative control). The white arrow represents the RPE layer; the yellow star is showing the photoreceptor outer segment layer; blue staining is DAPI.

3. Next, we tried staining RPE flatmounts with BODIPY 493/503. While there was a trend toward increased lipid accumulation in LCHADD RPE flatmounts, it was not statistically significant. Also, this test did not allow us to specifically identify the vacuoles seen in the RPE of retinal H&E cross-sections.

Figure 3: Images of RPE flatmounts from 15-month mice stained with BODIPY 493/503. Images taken at the 100X magnification of A) WT RPE flatmount and B) LCHADD RPE flatmount. C) Quantification of BODIPY fluorescence normalized to area of RPE. Images taken at 400X magnification of D) WT RPE flatmount and B) LCHADD RPE flatmount. F) Quantification of BODIPY fluorescence normalized to number of nuclei. Green is BODIPY 493/503; Blue is DAPI.

4. Finally, we tried staining bleached retinal cross-sections with ORO. Bleaching the retinal cross-sections allowed us to visualize staining in the RPE specifically. We, however, were not able to see a difference in ORO staining particularly in the RPE between the WT and LCHADD cross-sections, suggesting no increase in lipid droplets in RPE.

Figure 4: Image of retinal cross-sections from 12-month mice stained with oil red o after bleaching. Images taken at 600X magnification of A) WT and B) LCHADD mice. RPE is labeled by white arrow. Red is oil red o; blue is hematoxylin stain.

At this time, we have not identified the material in the RPE vacuoles seen on H&E cross-sections. We are continuing experiments in this effort but are hesitant to hold this manuscript based on this one point. We therefore changed the language to say RPE disruption or drop-out in the manuscript. Future planned experiments include: 1)staining retinal cross-sections with tunnel staining for dead cells, an 2)immune activation and infiltration.

- 1 Cox, K. B. *et al.* Gestational, pathologic and biochemical differences between very long-chain acyl-CoA dehydrogenase deficiency and long-chain acyl-CoA dehydrogenase deficiency in the mouse. *Human Molecular Genetics* **10**, 2069-2077 (2001).
- 2 Goetzman, E. S., Tian, L. & Wood, P. A. Differential induction of genes in liver and brown adipose tissue regulated by peroxisome proliferator-activated receptor-alpha during fasting and cold exposure in acyl-CoA dehydrogenase-deficient mice. *Mol Genet Metab* **84**, 39-47 (2005). [https://doi.org/S1096-7192\(04\)00250-1](https://doi.org/S1096-7192(04)00250-1) [pii]
10.1016/j.ymgme.2004.09.010
- 3 Tyni, T., Pihko, H. & Kivela, T. Ophthalmic pathology in long-chain 3-hydroxyacyl-CoA dehydrogenase deficiency caused by the G1528C mutation. *Curr Eye Res* **17**, 551-559 (1998).

REVIEWERS' COMMENTS:

Reviewer #1 (Remarks to the Author):

Authors have addressed my comments sufficiently for this report of the LCHADD mouse model. Future studies may be able to investigate some phenotypes in more detail.

Reviewer #2 (Remarks to the Author):

The authors did an excellent job revising this paper.

Reviewer #3 (Remarks to the Author):

The authors have addressed my concerns and have put a significant effort in doing so. I would like to thank them for their efforts. I have only one minor comment regarding the ERG traces on figure 6C. The time scales do not seem to match the traces. Typically the B-wave peaks is at around 100 msec and C-wave peak is between 500 and 1000 msec (see Pinto et al 2007, doi: 10.1007/s10633-007-9064-y). On figure 6C the B-wave is well below 50 msec and the C-wave is at about 100 msec. Based on the shape of the traces, it seems to me that an error was made in preparing the figure: the traces were compressed along the x-axis and the scales were not adjusted to match the trace. Other than that I would recommend the manuscript to be accepted for publication.